# Modeling Gastrointestinal Diseases Using Organoids to Understand Healing and Regenerative Processes

**DOI:** 10.3390/cells10061331

**Published:** 2021-05-27

**Authors:** Alexane Ollivier, Maxime M. Mahe, Géraldine Guasch

**Affiliations:** 1Aix-Marseille University, CNRS, INSERM, Institut Paoli-Calmettes, CRCM, Epithelial Stem Cells and Cancer Team, CEDEX 09, 13273 Marseille, France; alexane.ollivier@inserm.fr; 2Cincinnati Children’s Hospital Medical Center, Department of Pediatric General and Thoracic Surgery, Cincinnati, OH 45229, USA; maxime.mahe@cchmc.org; 3University of Cincinnati, Department of Pediatrics, Cincinnati, OH 45220, USA; 4UMR Inserm 1235-TENS, INSERM, Université de Nantes, Institut des Maladies de l’Appareil Digestif–CHU de Nantes, 1 Rue Gaston Veil, CEDEX 1, 44035 Nantes, France

**Keywords:** niche, homeostasis, organoid, gastrointestinal tract, metaplasia, inflammatory bowel diseases, disease-modeling, epithelial–mesenchymal crosstalk

## Abstract

The gastrointestinal tract is a continuous series of organs from the mouth to the esophagus, stomach, intestine and anus that allows digestion to occur. These organs are frequently associated with chronic stress and injury during life, subjecting these tissues to frequent regeneration and to the risk of developing disease-associated cancers. The possibility of generating human 3D culture systems, named organoids, that resemble histologically and functionally specific organs, has opened up potential applications in the analysis of the cellular and molecular mechanisms involved in epithelial wound healing and regenerative therapy. Here, we review how during normal development homeostasis takes place, and the role of the microenvironmental niche cells in the intestinal stem cell crypt as an example. Then, we introduce the notion of a perturbed niche during disease conditions affecting the esophageal–stomach junction and the colon, and describe the potential applications of organoid models in the analysis of human gastrointestinal disease mechanisms. Finally, we highlight the perspectives of organoid-based regenerative therapy to improve the repair of the epithelial barrier.

## 1. Introduction

Epithelia cover the body surface, such as the skin, and the internal parts such as the digestive system. These epithelial layers form barriers that protect us from various life assaults, e.g., pathogens, tissue damage or dysregulated immune response. During lifetimes, stem cells allow these epithelia to constantly self-renew and to differentiate into specialized cell types that give each tissue its specific function. The various cell types that compose the epithelium maintain a proper balance between cell proliferation and cell death during a process called homeostasis. Stem cells reside in a complex microenvironment called a niche, composed of diverse cell types such as fibroblasts, muscle cells, immune cells, endothelial cells, neural cells and adipocytes depending on the tissue [1,2,3]. Signals received by the niche dictate stem cell behaviors from developmental through adult stages. Certain organs in the gastrointestinal tract such as the esophagus, stomach, small intestine, colon, rectum, and anal canal, are frequently associated with chronic stress and injury during life such as acid reflux, irritation and tissue stretching, causing these tissues to be in frequent regeneration processes and thereby frequently subjected to the risk of developing disease-associated cancers. During these acute and chronic conditions, the crosstalk between niche cells and epithelial cells is often perturbed, resulting in the loss of the epithelial barrier [4].

Recently, the development of organoids from adult stem cells and from human pluripotent stem cells, allowing the formation of an organ in vitro that histologically and functionally resembles their tissue of origin, has allowed the study of the pathogenesis of these gastrointestinal diseases [5]. In this review we focus on advances made in organoid-based research in the gastrointestinal tract including the esophagus, stomach, small intestine and colorectal epithelia. We will first describe the importance of the intestinal niche in the establishment and maintenance of the epithelial homeostasis during development. These discoveries from the past decade underpinned the biological factors used to reproduce in organoid culture the in vivo niche components [6]. Then, we will introduce the notion of the perturbed niche in gastrointestinal diseases, and we will illustrate how the organoid-based technology can be used as a model of damaged tissue to understand the sequential steps of epithelial transformation and the molecular mechanisms involved in epithelial wound healing and regenerative therapy. Finally, we will highlight the perspectives of organoid-based regenerative therapy to improve the repair of the epithelial barrier.

## 2. Epithelial Cells and Their Niche: When Homeostasis Takes Place

The intestine is an excellent system to study homeostatic regulation and tissue morphogenesis because the spatial and temporal establishment of most lineages have been well defined [7]. Epithelial lineages derive from a common embryonic endodermal progenitor cell [8], and at adult stages undergo a constant turnover to maintain homeostasis and wounding. During development, an epithelial–mesenchymal crosstalk, involving FGF, WNT and CDX signaling, will establish position and outgrowth of the nascent villi in the intestinal tract [9] (Figure 1a). During late fetal development (e14 in mouse), the simple pseudostratified epithelium of the primitive gut tube, composed of undifferentiated cells, will become a simple glandular epithelium patterned into villi, and a continuous intervillous region that will later form the intestinal crypts (Figure 1b). In humans before birth, the final shape of the small intestine is characterized by crypt-villus units, in contrast to the adult colon that will lose the embryonic villi and contain only crypts. The transition from undifferentiated to mature intestinal epithelium involves all cells of the mouse epithelium, independently of their location or their expression of the intestinal stem cell marker, leucine-rich repeat-containing G-protein coupled receptor 5 (LGR5), a receptor for R-spondin molecules, suggesting that stem cell identity in that case is an induced property [10].

In addition to stem cells localized at the base of the intestinal crypt, transient amplifying (TA) cells also compose the epithelium. TA are progenitor cells that produce mature absorptive enterocytes, which make up the vast majority of the intestinal epithelial cells and secretory cells such as goblet, Paneth and enteroendocrine cells [11]. It is well known that the intestinal niche development is regulated by WNT, R-spondin and EGF signaling at the bottom of the crypt to promote cell replication while BMP signaling is highly expressed at the top of the villus to promote cell differentiation [12,13]. The spatial distribution of these signaling pathways comes from the variety of mesenchymal cells surrounding the intestine, with a defined role in the maintenance and the regeneration of the intestinal epithelium (Figure 1b). Along the crypt axis, there are sub-groups of stromal cells that have been characterized based on the expression of PDGFRα and other markers. PDGFRα^High^ cells, called telocytes, are predominantly found in the villus area and represent a source of BMP signals. In the intestinal subepithelium telocytes may play an essential role in modulating mechanical sensing such as the peristaltic movements in the gastrointestinal tract. They also express CD34, FOXL1, GLI1, SOX6, and CD90 [14]. Their role in the maintenance of the intestinal stem cell niche is crucial as deletion of FOXL1 positive telocytes impair the growth of the stem cells and progenitor intestinal cells resulting in the collapse of the crypts [15]. PDGFRα^Low^ cells expressing also CD81 are called trophocytes [16] and are in close contact to the intestinal stem cells at the base of the crypt. Trophocytes express the BMP antagonist Gremlin 1 that limits differentiation. These are crucial to sustain intestinal stem cell in vivo. Fibroblasts, expressing lower level of PDGFRα, and pericytes (PDGFRα negative), also compose the submucosa.

The establishment of the stem cell niche from development through adulthood is clearly dependent of the epithelial–mesenchymal crosstalk that is spatiotemporally controlled. Even though the role of each cell type in the gastro-intestinal mesenchyme is not yet fully understood, deregulation of this balance of signals may lead to preneoplastic condition.

## 3. Deregulated Niche and Epithelial Barrier Loss

Chronic inflammation, stress or injured conditions due to mechanical insults, chemical burns, or infection often affect the gastrointestinal tract, resulting in chronic diseases such as Barrett’s esophagus, gastric ulcer or Crohn’s disease affecting different areas of the digestive epithelium. Bowel disorders may be associated with defects in the surrounding epithelial niche, which has been less investigated.

The esophagus is an organ that carries food and liquids from the oral cavity to the stomach. It derives from the endoderm to form a simple columnar epithelium from esophageal epithelial precursors. Later during development, after the tracheal–esophageal separation, the epithelium becomes squamous, stratified with a basal layer of keratinocytes that contain proliferative and undifferentiated cells that will undergo differentiation within the overlying suprabasal cell layers. This multilayer epithelium is completely renewing within few weeks, and provides a barrier against mechanical trauma caused by food intake, acid reflux or pathogen colonization [17]. However, when the epithelium is subjected to repeat chronic inflammation, the proper balance between self-renewal, proliferation and differentiation is perturbed and can lead to the loss of its barrier protection. One example of perturbed homeostasis is the Barrett’s esophagus [18] where the normal stratified squamous epithelium of the esophagus is replaced by a metaplastic columnar epithelium, resembling intestinal epithelium with goblet cells and mucous layers in the lumen (Figure 2). This condition predisposes to esophageal adenocarcinoma. Ectopic expression of the homeobox gene *CDX2* at the squamo-columnar transition zone promotes intestinal metaplasia and a Barrett’s-like condition [19]. Interestingly, both bile and acid from the refluxed gastric liquid can induce expression of *CDX2* [20] suggesting that changes in the environment directly affects epithelial identity. Studies from human samples have shown the expression of the Hedgehog target genes *PTCH1* and *BMP4* in the Barrett stromal cells compared to normal condition [21]. Yet mesenchymal cells in Barrett’s esophagus have not been well characterized, and understanding which cell types in the microenvironment are directly impacted will be important to decipher the molecular mechanisms involved in the progression from normal to metaplasia condition.

The stomach is an organ that allows the secretion of acid to initiate digestion, and protects against foodborne pathogens. It is composed of various cell types including endocrine cells, parietal cells, chief cells, mucous gland neck cells and surface mucous pit cells. All of these cell types constitute the gastric mucosa that is continually exposed to environmental and physiological stress that can cause local epithelial damage [22]. Pathologies associated with the stomach include gastritis, gastric ulcers and gastric cancers. The mucosal surface is exposed to various microbes (bacteria and virus) [23]. When the lining of the stomach is wounded by open sores, gastric ulcers develop. The repair phases consist in a re-epithelization step, in which surrounding epithelial cells migrate and proliferate, followed by a regenerative phase where epithelial cells reform new continuous and functional glandular tissue [24]. Injuries to the gastric epithelium have two main causes: infection by *Helicobacter pylori* bacteria is the main cause of gastric inflammation and leads to gastritis, peptic ulcers and gastric cancer [25]. This pathogen colonizes the luminal side of the gastric epithelium by interacting with the apical side of the cell, causing an immune-inflammatory response on the stromal side of the mucosa. In addition, gastric acid stress, as demonstrated in mouse models, is associated with the expression of the pro-inflammatory mediator cyclooxygenage-2 (Cox-2), and acts as a co-promoting factor in early tumor formation [26], suggesting that, similarly to the esophagus, changes in the environment directly affects epithelial identity.

The colon, also called the large intestine, where most of the microbiota resides, is responsible for absorption of water and electrolytes, the chemical digestion by gut microbes, and the formation and transport of feces. Similar to the small intestine, the adult colon has crypts of Lieberkühn, where they harbor stem cells and Paneth cells in addition to endocrine cells found in their highest numbers. The stem cell and proliferating compartment in the colonic epithelium resides at the base of the crypt. In contrast to the small intestine, the colon has no villi. The differentiated cell types of the mature colon epithelium are the absorptive colonocytes and the goblet cells [27]. Mesenchymal niche cells surrounding the colon have been studied in humans, and reveal the presence of pericytes, myofibroblasts and a population of cells expressing *SOX6*, *CD142* and WNT genes crucial for stem cell maintenance and homeostasis [28]. The colonic tissue is commonly subjected to diseases affecting its homeostasis. Inflammatory bowel disease consists of two major idiopathic diseases: ulcerative colitis and Crohn’s disease, in which inflammation of the mucosa along the gastrointestinal tract is seen. During colitis, the mesenchymal niche is perturbed with the presence of an activated cell population expressing factors (such as IL-33, TNF superfamily member 14 and lysyl oxidases) participating in the inflammatory state, and epithelial barrier dysfunction [28]. These diseases are complex and multifactorial, with a polygenetic predisposition [29].

### Using Organoid to Model Damaged Epithelium and Wound-Healing Processes

Organoids are 3D structures made from adult or pluripotent stem cells, and display similar functional properties as the tissue of origin, including genetic and cellular heterogeneity. Stem cells are combined with the proper biological factors to reproduce in organoid culture the in vivo niche components (reviewed elsewhere [30]). Briefly, the media to grow intestinal and gastric organoids from adult stem cells contains Wnt3a, R-Spondin (interacts with Wnt), Noggin (inhibits BMP signaling), A83-01 (TGFb inhibitor), SB202190 (MAPK inhibitor), growth factors (EGF, FGF2, FGF7, FGF10), N2 supplement (human insulin, human transferrin, sodium selenite, putrescine, and progesterone), N-Acetyl Cysteine, Nicotinamide (vitamin B), gastrin (stimulates acid secretion and the cellular renewal of the epithelium) and the Rock inhibitor Y-27632 (to help recovering of the cells after their isolation). The utilization of organoids derived from both adult stem cells and human pluripotent stem cells offers new opportunities to understand epithelial homeostasis, in addition to physiopathological processes [31]. For example, intestinal organoids have been derived from adult [32] and induced pluripotent stem cells [33,34], and they contain not only all intestinal epithelial cell types, but also form crypts, microvilli and display transepithelial substance transporting capacity. Being able to maintain the mesenchymal niche cells and all epithelial cell types (such as tuft cells or enteroendocrine cells) in the intestinal organoid is crucial to studying disease states, as they are all implicated in inflammatory bowel disease [35,36].

Modeling damaged tissues by mimicking wounding processes at the organ level using both organoid technology and single cell genomics technologies will help in deciphering wound healing mechanisms at the cellular and molecular levels (Figure 3). Understanding these sequential steps will inform researchers about cell plasticity, and will define candidate genes associated with diseased conditions in addition to new therapeutics. A global effort has been led over the years to create faithful models of most of the organs of the gastro-intestinal tract [37].

A main advantage of using organoids is to bridge the gap between species. This has been exemplified with the esophagus. Human and mouse esophagi are histologically different, with several layers of basal cells and no keratinized cells in the human epithelium, in contrast to a single basal layer of cells and keratinization of the suprabasal layer in mice [17]. Due to these differences, it is therefore important to have reliable human in vitro models for research. Esophageal organoids have been derived from both adult [38] and human pluripotent cells [39] and provide a great tool to study metaplasia. Exogenous recombinant cytokines, such as interleukin-13, mimicking inflammatory conditions in eosinophilic esophagitis, has been used in esophagus organoid culture media, and relevant overexpressed genes have been monitored (*Sox2*, *Lox*, *CCL26*) to follow epithelial changes [38,40]. Culture from a Barrett’s esophagus epithelium has been successful in recreating the presence of alcian-blue, *TFF3* and *MUC2* positive goblet cells, however long-term survival of the culture was not optimal [41]. To study the malignant transformation of Barrett’s esophagus, genetically modified technology such as CRISPR-Cas9 genome editing of the APC gene has been used to constitutively activate WNT signaling [42]. APC^KO^ Barrett’s esophagus organoids display features of cancerous lesions including morphological dysplasia and hyperproliferation. To our knowledge, there is no report using organoids to study early steps of the esophageal epithelium transition to metaplasia. Nonetheless, metaplasia has been modeled in intestinal organoids derived from adult stem cells, where loss of Cdx2 induced a stomach-like epithelium [43]. Barrett’s metaplasia has been modeled in a p63 null mouse [44] that shows similarity in its gene expression profile when compared to human Barrett’s esophagus datasets. It will be interesting to identify genes associated with metaplasia initiation, such as p63, and create an inducible system to be used in normal esophageal organoids.

Gastric organoids had been first established from LGR5+ stomach adult stem cells [45], and later on from pluripotent stem cells [46]. These organoids contain all cell types normally found in the tissue, i.e., endocrine cells, parietal cells, chief cells, mucous gland neck cells and surface mucous pit cells. To mimic gastric wounds and to follow repair steps in organoids, it is crucial to integrate microbiota niches in the culture [37]. To that end, *Helicobacter pylori* can be microinjected into the lumen of the organoids and will adhere to the epithelium [46,47]. Organoids reproduce important hallmarks of the infection, as epithelial cells in gastric organoids secrete urea, which chemo-attract the bacteria and may contribute to their survival [48]. There are additional ways to mimic gastric wounds in organoids. Fundic gastric organoids have been used to study epithelial restitution using photodamage with a two-photon laser. The repair phase is then optically followed by measuring cell exfoliation and restoration of an intact epithelium caused by migration/proliferation of neighboring cells [49,50]. In vivo, an ulcer can be induced in mice by a localized application of acetic acid to the serosal surface of the stomach. It is therefore conceivable to acidify the culture media of organoids and follow epithelial changes (Figure 3). The gastric organoids were also used to study *Helicobacter pylori* mediated inflammation, and mechanisms of gastric carcinoma initiation and progression [51].

Organoid-based models of inflammatory bowel disease have been established from the inflamed mucosa of patients [52,53], and their characterization shows an epithelial alteration with some inflammatory features, such as pseudo-stratification, slow growth, altered polarization and decreased expression of tight-junction proteins. Organoids derived from colonic biopsies of inflammatory bowel disease patients have also been derived, and showed an upregulation of the pro-inflammatory cytokines, MCP-1 and TNF-a [54]. Interestingly, a pro-inflammatory cytokine cocktail (IL-1β, IL6, TNF-α) reproduced the inflammatory phenotype in non-inflammatory bowel disease organoids [55], suggesting that some aspect of these diseases can be induced in culture. Another important notion to consider is the compartmentalization of the gut where, depending of the location, immune cell composition may be different [55,56]. The organoid technology may provide a tool to investigate this aspect.

## 4. Organoid-Based Regenerative Therapy: Repairing the Epithelial Barrier

### 4.1. The Process of Regeneration Is A Healing Phase

In recent years, research work has shown promising therapy using organoids for patients with severe gastrointestinal epithelial injuries. A culture of organoid from LGR5 positive colon stem cells, and orthotopical transplantation experiments in acute colitis mice models, show functional engraftment in the affected area [57]. Transplanted cells integrate into the damaged tissue and differentiate properly to form newly functional crypts. Normal human colon organoids can reconstitute the colonic epithelium in a xenograft model, in which the murine colonic epithelium was disrupted using EDTA [58]. Ex-vivo cultured organoids from a murine fetal colonic epithelium has also shown regenerative capacity in a chemically induced colonic injury model [59]. Current methods to grow organoids use Matrigel, a tumor-derived extracellular matrix, that could not be used for regenerative and translational medicine in humans. To overcome this issue, human intestinal and colon organoids have been expanded in synthetically defined hydrogel [60]. Hydrogel-generated organoids differentiate properly in vivo and improve colonic wound healing. Because synthetic hydrogels are not derived from animal sources and are chemically defined with minimal batch-to-batch variability, their utilization is foreseen in clinical applications.

Organoid-based regenerative therapy also shows promise to address challenges in short bowel syndrome, a condition characterized by malabsorption and wasting conditions. Human small intestinal organoids can reconstitute the epithelium in a rat model of short bowel syndrome, having undergone a jejunoileal resection [61]. The resulting xenograft improves intestinal failure, a condition associated with diarrhea, dehydration and malnutrition, occurring in short bowel syndrome. Intestinal engineering has been also successful using human small intestine and colon scaffolds together with patient-derived intestinal organoids and fibroblasts to combat intestinal failure [62]. This work is a promising advance toward the clinical application of regenerative medicine using organoids to treat intestinal failure.

Organoid-based therapy has also been reported successfully for gastric tissue. The use of fluorescent gastric organoid and orthotopic transplantation into syngenic mice with ulcers shows that in aged stomachs, organoids promote healing [63]. When human gastric organoids are orthotopically transplanted into humanized mice with induced ulcers, they contribute to the regeneration of the gastric epithelium [64]. Therefore, endoscopy-assisted transplantation of organoids may not only help the regeneration of the refractory ulcers that may persist in inflammatory bowel diseases patients, but may also reduce the risk of developing colitis-associated cancers.

### 4.2. Role of the Stroma in the Regenerative Process

Wound healing is a complex process of biological and molecular events including angiogenesis, immune cell activation, tissue remodeling, cell migration and proliferation [65]. In the intestine, CD34+Gp38+ mesenchymal cells are part of the crypt stem cell niche and have been shown to participate in tissue repair by attracting inflammatory cells, secreting stem cell niche factors Grem1 and Rspo1, and upregulating growth factors for epithelial cells such as Amphiregulin, Fgf7, Fgf10, as well as Ptgs2, which modulates epithelial proliferation. All of these secretions may help to regenerate the intestinal epithelial barrier and restore homeostasis [66]. Human colonic mesenchyme has been profiled at the single cell level and reveals a distinct subset of cells involved in epithelial cell proliferation [28]. When compared to single cell analysis of mesenchymal cells in human ulcerative colitis, it highlighted dysregulation of some subsets of stromal cells enriched for TGFβ and BMP signaling, that could reflect the epithelial barrier dysfunction in this disease. Using chemically-challenged mice to mimic colitis, regenerative responses mediated by stromal cells have revealed two secreted factors of ulcerative colitis-associated stromal cells IL6 and TNFSF14. When added to organoid culture media, they induce stemness genes, suggesting a mechanism in which stem cells are recruited following inflammation mediated injury.

The stroma components are therefore an essential partner of epithelial cells in wound healing. In organoid-based regenerative therapy, organoids can be transplanted into a wounded area. It will be of interest to test if adding fresh non-inflamed stroma in combination with the organoids may improve stem cells engraftment and survival, opening new avenues for regenerative medicine.

## 5. Conclusions

The organoid technology has provided a perfect tool to get as close as possible to the tissue of origin because epithelial diversity and functions are maintained in culture. Growth factors given in the organoid culture media mimic the mesenchymal niche and allow epithelial cells in the organoid to survive long-term. However, not all cell types can be represented using this cocktail of growth factors, and the gradient of expression that exists in vivo cannot be reproduced in the culture yet. The size, shape and the polarity axis of the organoid are also a limitation because they do not reflect the reality of the organ. Topographically structured scaffolds offer a way to improve this in the future [67,68]. Studies from mice and humans have shown similarities, but also divergence in the composition of their niche in normal and diseased conditions. It is therefore crucial to improve the complex environment with matrices or organ-on-chip devices [69]. An inflammatory gut-on-chip model has allowed the study of the early inflammatory phase occurring in intestinal bowel disease. This system used the co-culture of the gut epithelium, the microbiome, and the immune cells and inflammation were induced by toxic chemical, bacterial toxin, or probiotic bacteria. This study shows that keeping an intact epithelial barrier function is a way to suppress inflammation in the human gut [70]. The recent technology allowing the profiling of tissue at the single cell level has not only highlighted the complexity of cellular and molecular regulators that maintain tissue homeostasis in the gastro-intestinal tract, but also brought forth the deregulated signals during chronic stress or injury. In conclusion, modeling gastro-intestinal disease in patients using organoids gives an opportunity to understand healing and regenerative processes, and may offer in the future new therapies using patient-derived organoids for transplantation.

## Figures and Tables

**Figure 1 cells-10-01331-f001:**
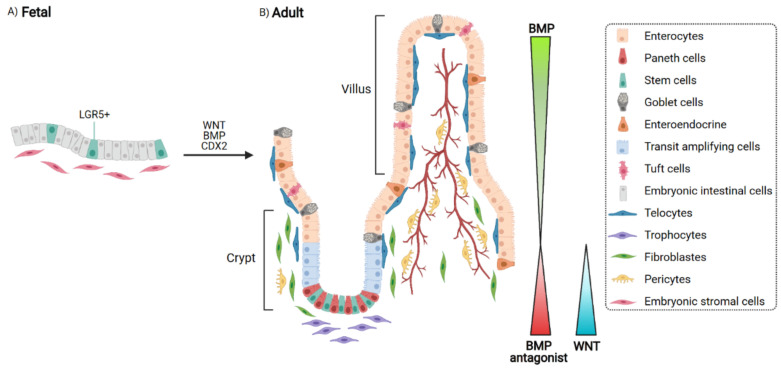
From fetal to adult stage: development of the intestinal stem cell niche. (**A**). At fetal age, the epithelium is pseudostratified and composed of cells expressing the stem cell marker LGR5. **(B**). At adult age, crypts and villi form and are surrounded by gradients of secreted molecules.

**Figure 2 cells-10-01331-f002:**
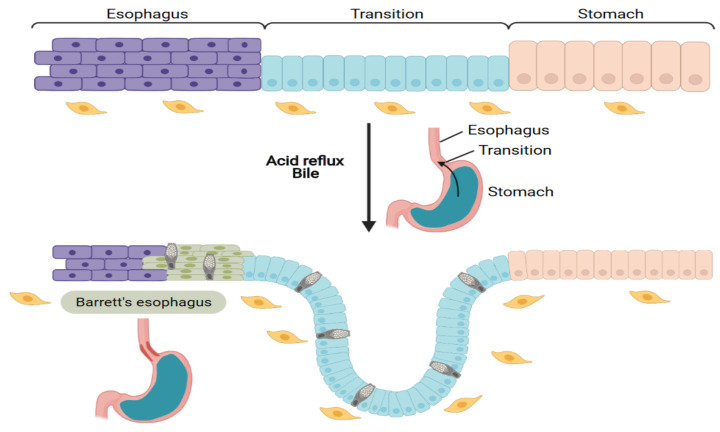
A metaplasic condition: the Barret’s esophagus. A transition zone exists between the stratified squamous epithelium of the esophagus and the glandular epithelium of the stomach. Under acid reflux, a metaplasia occurs and intestinal goblet cells are abnormally found in the epithelium.

**Figure 3 cells-10-01331-f003:**
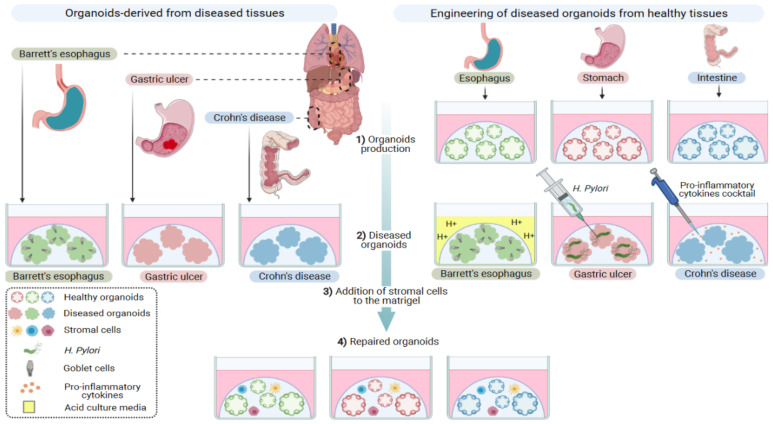
Organoid culture to understand the evolution of normal epithelia toward dysplasia and into neoplasia. Organoids can be derived from the esophagus, stomach and intestinal tissues from diseased or normal conditions. Mimicking disease states in culture could be considered by acidifying the culture media to mimic acid reflux in Barret’s esophagus, colonizing the gastric organoid with *H. pylori* or adding a pro-inflammatory cytokine cocktail to reproduce a Crohn’s disease condition. Co-culture of diseased organoids with stromal components could help decipher mechanisms involved in the repaired phase.

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
