# Peer review of "Modeling Gastrointestinal Diseases Using Organoids to Understand Healing and Regenerative Processes"

_cells, 2021, doi:10.3390/cells10061331_

Round 1

Reviewer 1 Report

This manuscript is well organized and straightforward on gastrointestinal organoids. That covers basic aspects in tissue/organ development to implying clinical applications such as regenerative therapy. Weighing descriptions on intestinal stem cell niche are relatively unique compared to other published reviews and could provide clues to study gastrointestinal biology and disease modeling for readers.

I have some minor comments.

Minor comments:

  1. Line 36; They should change “many cell types.” Many is an abstract word here.
  2. Line 125, 126; Somehow, bold characters were used. Please correct the bold.
  3. Line 125; “bile or acid”?
  4. Line 182-; The authors should cite papers to understand hPSC-derived intestinal organoids further instead of ref#32. A study of highly functional gastrointestinal organoids was ever reported below. That should be cited in this review.
  • Uchida H, Machida M, Miura T, Kawasaki T, Okazaki T, Sasaki K, Sakamoto S, Ohuchi N, Kasahara M, Umezawa A, Akutsu H. A xenogeneic-free system generating functional human gut organoids from pluripotent stem cells. JCI Insight. 2017; 2(1): e86492.
  1. Line 337-; References. The description format of cited papers is not unified. Also, we can find some typos, such as double periods.

Author Response

Reviewer 1 - Comments and Suggestions for Authors

This manuscript is well organized and straightforward on gastrointestinal organoids. That covers basic aspects in tissue/organ development to implying clinical applications such as regenerative therapy. Weighing descriptions on intestinal stem cell niche are relatively unique compared to other published reviews and could provide clues to study gastrointestinal biology and disease modeling for readers.

I have some minor comments.

Line 36; They should change “many cell types.” Many is an abstract word here.

We have removed many cell types and replaced by ‘’diverse cell types such as fibroblasts, muscle cells, immune cells, endothelial cells, neural cells and adipocytes depending on the tissue’’

Line 125, 126; Somehow, bold characters were used. Please correct the bold.

There were no bold characters in our initial submitted version of the manuscript in word version. This is probably coming from the editorial office

Line 127; “bile or acid”?

We have modified this point as both bile and acid from the refluxed gastric liquid can induce the expression of CDX2.

Line 182-; The authors should cite papers to understand hPSC-derived intestinal organoids further instead of ref#32. A study of highly functional gastrointestinal organoids was ever reported below. That should be cited in this review.

We have added this reference as suggested. This is now reference 34

Uchida H, Machida M, Miura T, Kawasaki T, Okazaki T, Sasaki K, Sakamoto S, Ohuchi N, Kasahara M, Umezawa A, Akutsu H. A xenogeneic-free system generating functional human gut organoids from pluripotent stem cells. JCI Insight. 2017; 2(1): e86492. 

Line 337-; References. The description format of cited papers is not unified. Also, we can find some typos, such as double periods.

We have corrected all typos in the reference list

Reviewer 2 Report

Ollivier et al presented a review entitled "Modeling Gastrointestinal Diseases using Organoids to Understand Healing and Regenerative Processes". The review includes the followings:  a)Epithelial cells and their niche: when homeostasis takes place.   b)Deregulated niche and epithelial barrier loss. c)  Using organoid to model damaged epithelium and wound-healing processes. d)Organoid-based regenerative therapy: repairing the epithelial barrier which includes the process of regeneration is a healing phase,  and role of the stroma in the regenerative process, 

I guess the review is missing several point regarding the organoids growth and application

a) What is the basic composition required for organoid growth? The authors mentioned briefly WNT, R-spondin. However, other compnents such as Noggin, EGF, nicotinamide, gastrin, ROCK inhibitors ,...etc are required. Please mention them and their functions.

b) What is the role of organoid in studying GIT diseases? The authors mentioned briefly that oragnoids were isolated from IBD. However, the authors should show the role of organoids in studying other GIT diseases such as gastric cancer, colon cancer,..etc.

c) The authors mentioned the advantage of organoids? What are the disadvantges for this model?

Author Response

Reviewer 2 - Comments and Suggestions for Authors

Ollivier et al presented a review entitled "Modeling Gastrointestinal Diseases using Organoids to Understand Healing and Regenerative Processes". The review includes the followings:  a)Epithelial cells and their niche: when homeostasis takes place.   b)Deregulated niche and epithelial barrier loss. c)  Using organoid to model damaged epithelium and wound-healing processes. d)Organoid-based regenerative therapy: repairing the epithelial barrier which includes the process of regeneration is a healing phase,  and role of the stroma in the regenerative process, 

I guess the review is missing several point regarding the organoids growth and application

  1. a) What is the basic composition required for organoid growth? The authors mentioned briefly WNT, R-spondin. However, other components such as Noggin, EGF, nicotinamide, gastrin, ROCK inhibitors ,...etc are required. Please mention them and their functions.

In the section 3.1 Line 180 we have added this sentence: Stem cells are combined with the proper biological factors to reproduce in organoid culture the in vivo niche components (reviewed elsewhere30). Briefly, the media to grow intestinal and gastric organoids from adult stem cells contains Wnt3a, R-Spondin (interacts with Wnt), Noggin (inhibits BMP signaling), A83-01 (TGFb inhibitor), SB202190 (MAPK inhibitor), growth factors (EGF, FGF2, FGF7, FGF10), N2 supplement (human insulin, human transferrin, sodium selenite, putrescine, and progesterone), N-Acetyl Cysteine, Nicotinamide (vitamin B), gastrin (stimulates acid secretion and the cellular renewal of the epithelium) and the Rock inhibitor Y-27632 (to help recovering of the cells after their isolation).

  1. b) What is the role of organoid in studying GIT diseases? The authors mentioned briefly that organoids were isolated from IBD. However, the authors should show the role of organoids in studying other GIT diseases such as gastric cancer, colon cancer,..etc.

In this review we wanted to focus on the role of organoid to study the early stage of disease affecting the gastro-intestinal tract and as shown in the figure 3 the organoid can be used to model the evolution of normal epithelia toward dysplasia and into neoplasia. Our review bring this novel aspect and we think that showing the role of organoids in studying GIT cancers will unfocused the review.

  1. c) The authors mentioned the advantage of organoids? What are the disadvantges for this model?

We have added a sentence of some disadvantages of using organoids in the conclusion

‘’ However, not all cell type can be represented using this cocktail of growth factors and the gradient of expression that exists in vivo cannot be reproduced in the culture yet. The size, shape and the polarity axis of the organoid are also a limitation because they do not reflect the reality of the organ. Topographically structured scaffolds offer a way to improve this in the future’’. We have added two new references (#66-67).

Reviewer 3 Report

The manuscript entitled “Modeling Gastrointestinal Diseases using Organoids to Understand Healing and Regenerative Processes” seems to be covered a majority of the topics. The reference list covers relevant literature reflecting the recent developments made in this research area. However, some grammatical errors need to be rectified throughout the manuscript.

Author Response

Reviewer 3

The manuscript entitled “Modeling Gastrointestinal Diseases using Organoids to Understand Healing and Regenerative Processes” seems to be covered a majority of the topics. The reference list covers relevant literature reflecting the recent developments made in this research area. However, some grammatical errors need to be rectified throughout the manuscript.

We have carefully corrected all grammatical errors throughout the manuscript.

Round 2

Reviewer 2 Report

The authors replied properly to my questions. Thank you for this.

I have minor suggestions 

In line 245 page 6: Please enumerate the gastric organoids were used to study H.pylori mediated inflammation and mechanisms of gastric carcinoma initiation and progression . Please cite this reference PMID: 32518160

line 246-252: organoids isolated from IBD showed that upregulation of MCP-1 and TNFa in the IBD patients and the crosstalk between epithelium cells and immune cells is one cause of inflammation associated with IBD.  PMID: 32003126

Author Response

Reviewer 2 - Comments and Suggestions for Authors

I have minor suggestions 

In line 245 page 6: Please enumerate the gastric organoids were used to study H.pylori mediated inflammation and mechanisms of gastric carcinoma initiation and progression . Please cite this reference PMID: 32518160

As suggested by the reviewer we have added this sentence in the gastric organoid section and add the reference below, now reference #52:

Sayed IM, Sahan AZ, Venkova T, Chakraborty A, Mukhopadhyay D, Bimczok D, Beswick EJ, Reyes VE, Pinchuk I, Sahoo D, Ghosh P, Hazra TK, Das S. Helicobacter pylori infection downregulates the DNA glycosylase NEIL2, resulting in increased genome damage and inflammation in gastric epithelial cells. J Biol Chem. 2020 Aug 7;295(32):11082-11098.

line 246-252: organoids isolated from IBD showed that upregulation of MCP-1 and TNFa in the IBD patients and the crosstalk between epithelium cells and immune cells is one cause of inflammation associated with IBD.  PMID: 32003126

‘’We have added this sentence ‘’ Organoids derived from colonic biopsies of inflammatory bowel disease patients have also been derived and showed an upregulation of the pro- inflammatory cytokines, MCP-1 and TNF-a ’’ and added the reference below, now reference # 55.

Sayed IM, Suarez K, Lim E, Singh S, Pereira M, Ibeawuchi SR, Katkar G, Dunkel Y, Mittal Y, Chattopadhyay R, Guma M, Boland BS, Dulai PS, Sandborn WJ, Ghosh P, Das S. Host engulfment pathway controls inflammation in inflammatory bowel disease. FEBS J. 2020 Sep;287(18):3967-3988.